# Complicated Long Term Vaccine Induced Thrombotic Immune Thrombocytopenia—A Case Report

**DOI:** 10.3390/vaccines9111344

**Published:** 2021-11-17

**Authors:** Albrecht Günther, Dirk Brämer, Mathias W. Pletz, Thomas Kamradt, Sabine Baumgart, Thomas E. Mayer, Michael Baier, Angelina Autsch, Christian Mawrin, Linda Schönborn, Andreas Greinacher, Thomas Thiele

**Affiliations:** 1Hans-Berger-Department of Neurology, Jena University Hospital, 07743 Jena, Germany; dirk.braemer@med.uni-jena.de; 2Institute for Infectious Diseases and Infection Control, Jena University Hospital, 07743 Jena, Germany; Mathias.pletz@med.uni-jena.de; 3Core Facility Cytometry, Institute for Immunology, Jena University Hospital, 07743 Jena, Germany; thomas.kamradt@med.uni-jena.de (T.K.); sabine.baumgart@med.uni-jena.de (S.B.); 4Department of Neuroradiology, Jena University Hospital, 07743 Jena, Germany; t.e.mayer@med.uni-jena.de; 5Institute for Medical Microbiology, Jena University Hospital, 07743 Jena, Germany; michael.baier@med.uni-jena.de; 6Forensic Medicine Department, Jena University Hospital, 07743 Jena, Germany; angelina.autsch@med.uni-jena.de (A.A.); christian.mawrin@med.ovgu.de (C.M.); 7Department of Neuropathology, University Hospital Magdeburg, 39120 Magdeburg, Germany; 8Department for Transfusion Medicine, Institute for Immunology and Transfusion Medicine, University Medicine Greifswald, 17487 Greifswald, Germany; linda.schoenborn@med.uni-greifswald.de (L.S.); andreas.greinacher@med.uni-greifswald.de (A.G.); thomas.thiele@med.uni-greifswald.de (T.T.)

**Keywords:** vaccine, COVID-19, thrombocytopenia, cerebral thrombosis, autoimmune, case report

## Abstract

Background and Objectives: Vaccine induced thrombotic thrombocytopenia (VITT) may occur after COVID-19 vaccination with recombinant adenoviral vector-based vaccines. VITT can present as cerebral sinus and venous thrombosis (CSVT), often complicated by intracranial hemorrhage. Today it is unclear, how long symptomatic VITT can persist. Here, we report the complicated long-term course of a VITT patient with extremely high titers of pathogenic anti-platelet factor 4 (PF4)-IgG antibodies. Methods: Clinical and laboratory findings are presented, including the course of platelet counts, D-Dimer levels, clinical presentation, imaging, SARS-CoV-2-serological and immunological, platelet activating anti-PF4-IgG, as well as autopsy findings. Results: The patient presented with extended superior sagittal sinus thrombosis with accompanying bifrontal intracerebral hemorrhage. Repeated treatment with intravenous immune globuline (IVIG) resolved recurrent episodes of thrombocytopenia. Moreover, the patient’s serum remained strongly positive for platelet-activating anti-PF4-IgG over three months. After a period of clinical stabilization, the patient suffered a recurrent and fatal intracranial hemorrhage. Conclusions: Complicated VITT with extremely high anti-PF4-IgG titers over three months can induce recurrent thrombocytopenia despite treatment with IVIG and anticoagulation. Plasma exchange, immunoadsorption, and /or immunosuppressive treatment may be considered in complicated VITT to reduce extraordinarily high levels of anti-PF4-IgG. Long-term therapy in such cases must take the individual bleeding risk and CSVT risk into account.

## 1. Introduction

Vaccine-induced immune thrombosis with thrombocytopenia (VITT) is an adverse event that may occur after vaccination with vector-based ChAdOx1 nCOV-19 (AstraZeneca, Nijmegen, The Netherlands) or AD26.COV2S (Janssen Vaccines & Prevention B.V., Leiden, The Netherlands) [1,2]. In addition, the Brighton collaboration introduced the term—thrombosis with thrombocytopenia syndrome [3].

Typical symptoms of VITT are thrombocytopenia, thrombosis, and disseminated intravascular coagulation beginning in a time window of 4 to 20 days post-vaccination [1,4,5]. Thromboses include unusual venous or arterial thromboembolic events, especially cerebral venous and sinus thrombosis (CVST), very high D-Dimer levels [6,7]. Further, VITT—patients can present with severe headaches in this time window in the absence of thrombosis. This “pre-VITT syndrome” was described, including headache as the major symptom [8]. VITT is caused by IgG antibodies directed against platelet factor 4 (PF4), which activate platelets via their FcgRIIa receptors [1]. Their detection in conjunction with the clinical symptoms confirms the diagnosis of VITT.

First-line treatment of VITT involves the immediate start of high dose intravenous immune globulin (IVIG) and non-heparin anticoagulation [9]. Furthermore, immunosuppression and plasma exchange have been applied to treat severe VITT, when patients had persisting thrombocytopenia and elevated D-dimers despite first-line treatment [10].

The nature of the immune reaction behind VITT is currently debated. VITT shares close similarities to autoimmune heparin-induced thrombocytopenia and likely follows a two-step mechanism. First, PF4 forms a neo-epitope together with constituents of vector vaccines [11]. This results in a specific secondary immune response producing high levels of anti-PF4-IgG. How far epitope similarities between PF4 and the spike-protein of SARS-CoV-2 contribute remains controversial. Members of our group did not find any cross-reactivity between VITT antibodies and the SARS-CoV-2 spike-protein [12].

Platelet-activating anti-PF4 IgG antibodies decrease over time in VITT [13] in the majority of patients thus that VITT appears to be self-limiting. However, some patients continue to have high antibody titers confronting physicians with the question, whether VITT could become a chronic disease with persisting high anti-PF4-IgG levels. This may have consequences for treatment, especially for the duration of anticoagulation and/or the necessity of additional immunosuppression.

Here, we report the complicated long-term course of a patient with VITT and describe laboratory, immunological, brain imaging, and autoptic features and discuss potential implications for diagnosis and treatment of VITT.

## 2. Case Report

A 54-year-old Caucasian male presented with new onset of severe headache 12 days after first vaccination with ChAdOx1 nCoV-19. His prior medical history was completely unremarkable without evidence for an increased thrombotic risk. The patient did not receive any regular medication. On admission, he had thrombocytopenia and signs of disseminated intravascular coagulation (platelet count 37 × 10^9^/L; fibrinogen 0.94 g/L; D-dimer > 30 mg/L; Figure 1). SARS-CoV-2 infection was ruled out by PCR from nasopharyngeal swab. Serological antibody testing revealed a normal SARS-CoV-2-spike IgG titer without a signal to nucleocapsid in chemiluminescent immunoassay (CLIA) and immunoblot testing compatible with a vaccine-induced, but not SARS-CoV-2-infection related immunological response (at day 18 after vaccination). Cranial magnetic resonance imaging showed extensive superior sagittal sinus vein thrombosis with frontal hemorrhagic infarction (Figure 2). Thoracic, abdominal, and pelvic computed tomography (CT) revealed bilateral adrenal hemorrhages (not shown). Follow-up cranial CT revealed CVST with intracranial hemorrhage (ICH) progression to both frontal lobes, requiring hemicraniectomy within 24 h after admission. IVIG (1 g/kg body weight on two consecutive days each) and anticoagulation with argatroban were started immediately. Platelet count increased to 282 × 10^9^/L until day 5. However, three additional courses of IVIG were required to resolve recurrent, almost weekly occurring episodes of thrombocytopenia. Anticoagulation was changed from argatroban to tinzaparine, to argatroban, to danaparoid, and finally to fondaparinux. Anticoagulation was switched four times because of pharmacologic considerations (liver toxicity, half-life, possibility of antagonization) without affecting the platelet count course (Figure 1B). After ruling out cross-reactivity of VITT antibodies with PF4/heparin complexes, heparin was started, which was well tolerated. Due to recurrent episodes of thrombocytopenia, immunosuppression with prednisolone (70 mg orally) was initiated. From thereon, platelets remained >150 × 10^9^/L, but still dropped from 581 to 182 × 10^9^/L after three months, which could not be explained by changes in medication or other new co-morbidities. During intensive care unit (ICU) treatment, the patient suffered recurrent epileptic seizures requiring antiepileptic treatment, pneumonia with the need for antibiotic treatment but remained otherwise clinically stable. The patient was transferred to a neurological rehabilitation center 40 days after admission and returned to our hospital after 6 weeks for diagnostic and therapeutic re-evaluation. The minor recurrent ICH (Figure 2V) was diagnosed in a routine imaging control without a clinical correlate at this time point. He was sent to another rehabilitation center at day 90 for further neurological rehabilitation. On day 103 he developed new clinical signs for increased intracranial pressure with vomiting, decreased level of consciousness, pupillary abnormalities, and clinical swelling at the site of hemicraniectomy as a sign of herniation. Four months after VITT onset, the patient developed a new space-occupying intracerebral hemorrhage as a secondary complication to previous CVST/ICH and anticoagulation (Figure 2V,VI). According to the patient’s and family’s wish, ICU treatment goal was changed to a comfort care approach, and the patient deceased four days later.

## 3. Brain Imaging

Several cranial brain imaging studies were performed during the course of the patient’s hospital stay using computed tomography and magnetic resonance tomography accompanied with contrast application for venography.

Cranial plain computed tomography (CT) and magnetic resonance imaging (MRI) initially showed thrombosis of the rostral part of the superior sagittal sinus with left frontal hemorrhagic infarction (Figure 2I,II).

MRI after decompression hemicraniectomy revealed no new pathology (Figure 2III; right frontal ICH was detected on pre-operative CT scan already). In addition, during the following weeks, imaging detected recanalization of the superior sagittal sinus with no further cerebral injury (Figure 2IV). Three months after initial hemorrhage, a new small left temporal ICH was present (Figure 2V), and after four months, a massive hemorrhage in the left parietal and temporal lobe occurred (Figure 2VI) without signs of new thrombosis.

## 4. Laboratory Investigations 

### 4.1. Anti-PF4/Polyanion Antibody Testing

An in-house IgG-specific PF4/polyanion EIA [14] was used to screen for antibodies recognizing PF4 and PF4/heparin complexes [15]. Positive results were given in optical density (OD) units, as follows (reference range < 0.50): weak reaction, 0.5 to ≤1.0 units; strong reaction > 1.0 units). Samples were further assessed for platelet-activating antibodies in a washed platelet activation assay in the presence of PF4 (10 µg/mL) [16]. 

High titers of platelet-activating anti-PF4 IgG were detectable throughout the entire course, indicating ongoing VITT for more than three months. The optical densities of PF4/heparin ELISA at admission and day 90 after symptom onset were 3.05/2.98, respectively (tested as described in detail [1]). Both sera demonstrated platelet activation in a washed platelet assay in the presence of PF4, which is the typical laboratory pattern of VITT [1].

### 4.2. SARS-CoV-2 Antibody Testing

All samples underwent testing for specific SARS-CoV-2 IgG-antibodies with the following assays: LIAISON^®^ SARS-CoV-2 S1/S2 IgG on Liaison XL instrument (DiaSorin, Sallugia, Italy) deploying a recombinant antigen from the S1 and S2 domain of SARS-CoV-2, Elecsys Anti-SARS-CoV-2 on cobas e801 (Roche, Mannheim, Germany) using a recombinant nucleocapsid protein as capture antigen and hence identifying only infection associated antibodies, and recomLine SARS-CoV-2 IgG (Mikrogen, Neuried, Germany) as a line blot assay providing the S1- (spike), RBD- (receptor binding domain) and NP- (nucleoprotein) antigen of SARS-CoV-2 as well as the specific NP antigens of all 4 seasonal coronaviruses 229E, NL63, OC43, HKU1. All tests were performed following the manufacturer’s instructions. The tests performed were chosen corresponding to the appropriate benchmark reference [17].

There were three sequential serological test points before initiation of IVIG treatment and on days 5 and 18 after disease onset revealing a S1/S2-Antigen-CLIA signal of 21.8, 69.2, and 39.7 AU/mL and corresponding signals in the line blot assay. However, no N-Antigen signal was detectable.

We also tested IVIG-probes in order to rule out contamination with iatrogenic antibodies. The specific antibody content in undiluted IVIGs showed similar or lower values (25.8, 29.6, and 48.9 AU/mL) in comparison to the patient’s samples. In one IVIG solution, a positive signal for the infection-associated N antigen was detected but never in the patient’s samples. Hence, after administration, IVIGs did not essentially contribute to the measured patient’s SARS-CoV-2-specific IgG titers.

Additionally, signals against all four seasonal coronaviruses (CoV) were detected in all three serial patient samples with a higher semi-quantitative result for the other than SARS-CoV-2 beta-CoV (OC43 and HKU1, respectively), but no significant change in intensity over the observation period was detectable.

### 4.3. Immunophenotyping

Whole blood samples were processed according to the standardized protocol of the MAXPAR Direct Immune Profiling Assay provided by Fluidigm (South San Francisco, CA, USA) and measured on a Helios instrument (CyTOF, Fluidigm). Data were further analyzed by using Pathsetter software (Fluidigm).

Multidimensional mass cytometric single cell analyses [18] of whole blood samples on day 5, 21, 35, 75 after disease onset revealed lymphopenia and increased frequencies of granulocytes. Compared to day 5 there were striking changes with increased (CD4^+^ effector memory cells, Th1- and Th2-, and regulatory T cells) or decreased naïve T cells, B cells, and eosinophils frequencies in many leukocyte subsets at day 21 (Figure 3). Particularly at day 21 there was a strong increase in the percentage of TCRχ*δ* T cells (Figure 4). All these differences became diminished at day 35 except frequencies of monocytes (nadir at day 35) and plasmablasts (maximal frequency at day 35, Figure 4). By day 75, all frequencies had returned to baseline level. Across all time points, frequencies of naïve CD8^+^ T cells and TCRχ*δ* T cells were increased in comparison to age-matched healthy male donors.

## 5. Forensic Obduction

Postmortem autopsy was performed in accordance to the legal regularities. On autopsy, a residual thrombus in the left sinus transversus without evidence for other thromboembolic pathology in the brain or other solid organs was found. The brain showed signs of severe edema. Several hemorrhages were detectable predominantly in the left hemisphere. Microscopy revealed large hemorrhages, as well as small perivascular hemorrhages and extensive neuronal death together with brain edema (as shown in Figure 5). 

Apart from this, a florid bronchopneumonia and a small liver hemangioma were diagnosed. As the cause of death, the new left-sided, temporal-parietal space-occupying intracerebral hemorrhage was seen.

## 6. Discussion

The patient presented with typical symptoms of VITT, including thrombocytopenia, CSVT, signs of DIC. He also developed adrenal hemorrhage, a condition seen in other anti-PF4 IgG-mediated diseases such as heparin-induced thrombocytopenia [19]. He further developed a complicated course of VITT, including a rapid neurological and neuroradiological progress within 24 h after VITT diagnosis. VITT often involves very rapid progress of brain edema and ICH [20], which contrasts to non-VITT CSVT where massive brain edema and hemicraniectomy occurs less frequently [21].

The complicated long-term course of the present case has to be seen in conjunction with high titers of platelet-activating anti-PF4 IgG persisting for longer than three months. During the early course, recurrent episodes of thrombocytopenia developed as a consequence of platelet activation and platelet consumption. This was supported by the concomitant increase of D-Dimers during recurrent thrombocytopenia. High titers of anti-PF4-IgG platelet activating antibodies promote the prothrombotic condition in VITT and therapeutic dose anticoagulation is necessary to reduce the risk of thrombosis. We consider it likely that the prothrombotic state returns when anticoagulation is stopped albeit platelet-activating antibodies remain at high titers, based on our experience with autoimmune heparin-induced thrombocytopenia [22]. However, the ICH developed as a complication from lesions acquired early in VITT-related CSVT and secondary to anticoagulation. It is, therefore, important to define the optimal time point and strategy to stop or reduce the dose of anticoagulation in VITT (e.g., by close monitoring of platelet counts, D-Dimers, and platelet-activating anti-PF4-IgG).

These observations have major implications for VITT treatment. First, early repeated decreases of platelet counts can successfully be treated with repeated high doses of IVIG combined with therapeutic dose anticoagulation. Second, these patients may require prolonged therapeutic dose anticoagulation exceeding three months of treatment. Although pathogenic anti-PF4 IgG causing VITT typically decreases [13], VITT can induce extremely high levels of anti-PF4-IgG. Given a half live for IgG of 21 days, extremely high titers produced at the onset of VITT can remain for longer periods exceeding three months. This is associated with a high risk of thrombosis, therefore, repeated testing for anti-PF4-IgG is indicated in patients with ongoing symptoms or recurrent thrombocytopenia. Third, it may be appropriate to decrease anti-PF4-IgG titers by plasma exchange or even more effectively by immuno-adsorption very early in patients with severe VITT, who are refractory to first-line treatment with IVIG and non-heparin anticoagulation [10]. On the other hand, other antiplatelet antibodies may increase the bleeding risk when thrombocytopenia worsens. This needs to be balanced against the risk of thrombosis when the decision for anticoagulation is made. Thrombocytopenia recovered significantly after starting prednisolone, a treatment that was initiated to decrease the production of platelet-activating anti-PF4-IgG. However, in retrospect, a therapy with rituximab could have been beneficial to mitigate the massive production of anti-PF4-IgG by a “hyper-active” B-cell clone at the onset of VITT and to provide early treatment for concomitant autoimmune thrombocytopenia.

Besides the detection of platelet-activating anti-PF4-IgG, the exclusion of a COVID-19 infection is important since a relevant number of patients develop COVID-19 infection within a few days from vaccination. While we could rule out acute SARS-CoV-2 infection, we found that IVIG-products contain SARS-CoV-2 N-Antigen-IgG, which was never found in the patient’s serum probes. These antibodies against seasonal CoV such as OC43 and HKU1 are attributed to mitigate deterioration of COVID-19 [23,24]. If antibodies against the spike domain of seasonal HcoV contribute to the pathogenesis of VITT needs to be further evaluated [25]. We conclude that testing of any SARS-CoV-2 antibodies should be performed from a blood sample obtained before IVIG administration to avoid misinterpretations.

Deep profiling of leukocytes by mass cytometry revealed massively increased numbers and frequencies of gamma/delta T-lymphocytes and other T cell subsets. This observation may be related to pneumonia because peak numbers of these cells were detected four days after CRP levels increased (data not shown) [26]. Moreover, the frequency of plasmablasts, indicative of massive ongoing antibody production, was strongly increased together with reduced frequencies of monocytes. However, deep profiling of leukocytes by mass cytometry revealed that the composition of leukocyte subpopulations normalized over time compared to a healthy donor cohort, suggesting ongoing recovery of the immune system. As these data are preliminary, we cannot conclude on any VITT or CVST specific aspects. This will need to be established in a larger cohort of VITT patients.

## 7. Conclusions

We conclude that high titers of platelet-activating anti-PF4 antibodies may persist for more than three months in VITT. Close monitoring of platelet counts, D-Dimers, and anti-PF4 IgG and their platelet-activating potential is necessary to guide treatment decisions in patients with severe and long-term VITT.

## Figures and Tables

**Figure 1 vaccines-09-01344-f001:**
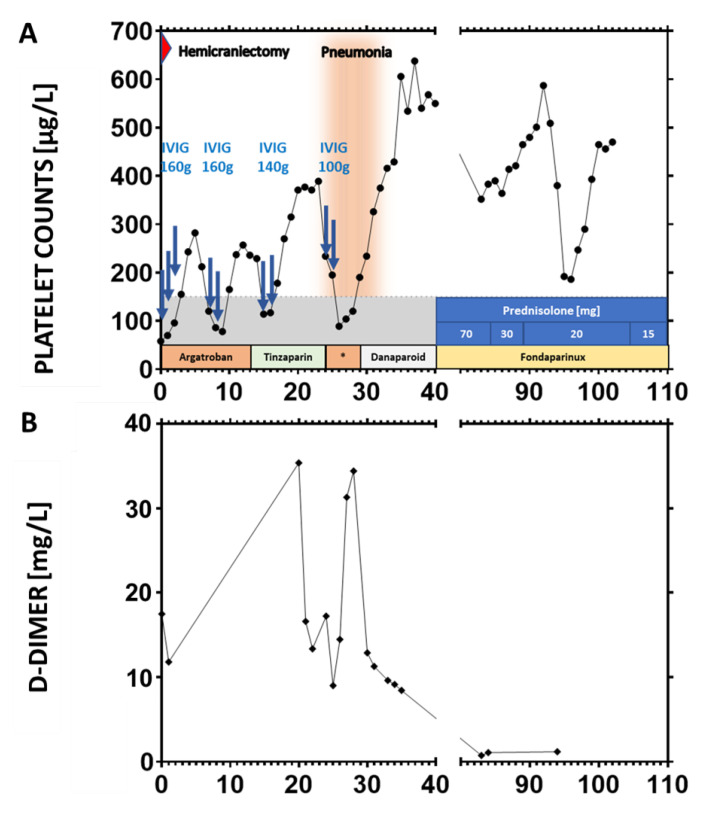
Course of Platelet Counts, anticoagulation treatment, and D-Dimers. Recurrent thrombocytopenia (**A**) and D-Dimer elevations (**B**) in perspective to disease onset (in days) on the x-axis. The grey area indicates the platelet range for thrombocytopenia. Intravenous immune globuline (IVIG) doses refer to the total doses (~2 g/kg body weight) administered at two or three consecutive days (each arrow indicates an administration). Dosages of anticoagulants were as follows: argatroban: aPTT controlled administration with changing dosage (aiming at aPTT of 60 s; approximately 2 µg/kg/min); tinzaparin: 12.000 U/day s.c.; danaparoid: 750 U s.c. × 3 times/d according to anti-Xa-Level; fondaparinux: 7.5 mg/d s.c.; Fondaparinux was administered until the treatment goal was changed to best supportive care after diagnosis of malignant recurrent ICH.

**Figure 2 vaccines-09-01344-f002:**
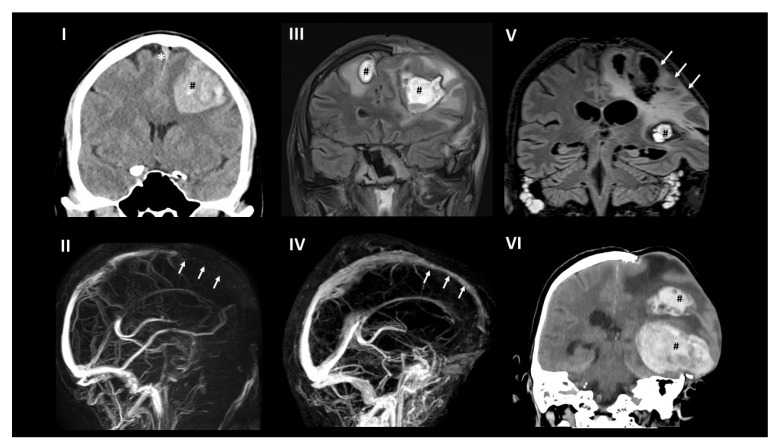
Brain imaging. (**I**) (day 1): In plain CT superior sagittal sinus thrombosis (*) and left frontal hematoma (#) was present. (**II**) (day 1): contrast-enhanced MRI revealed extensive superior sagittal sinus thrombosis (arrows). (**III**) (day 21): MRI (fluid-attenuated inversion recovery; FLAIR) three weeks after decompression hemicraniectomy showed the known bifrontal hemorrhagic infarction (#). (**IV**) (day 87): contrast-enhanced MRI demonstrated recanalization of the superior sagittal sinus (arrows). (**V**) (day 87): MRI FLAIR revealed a new small ICH in the temporal lobe (#) and the chronic lesions (arrows) with significant prolapse of the left cerebral hemisphere. (**VI**) (day 103): plain CT demonstrated fatal new left parietal and temporal ICH (#).

**Figure 3 vaccines-09-01344-f003:**
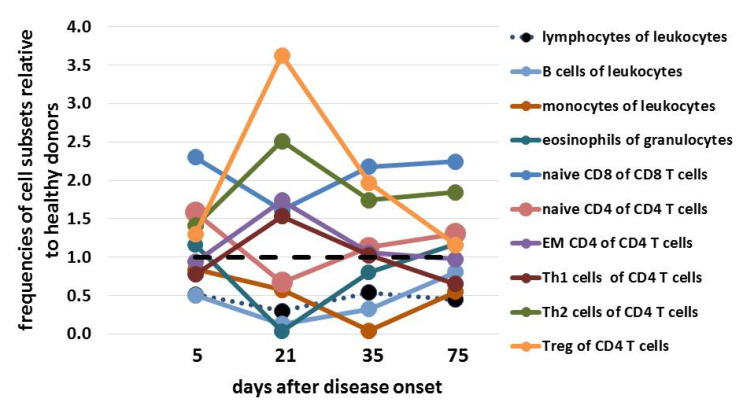
CyTOF results showing course of differential immunological cell subsets.

**Figure 4 vaccines-09-01344-f004:**
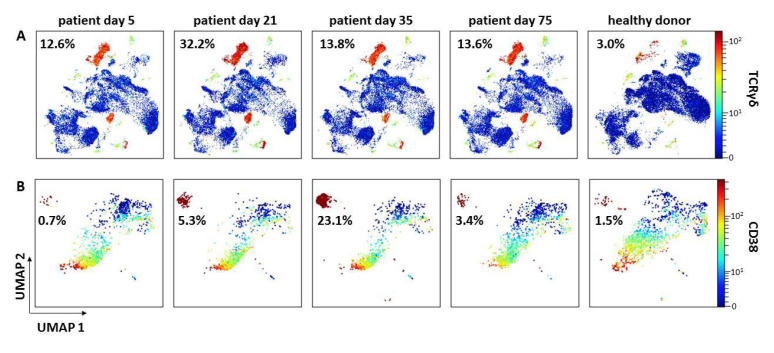
Uniform Manifold Approximation and Projection for Dimension Reduction (UMAP) of pre-gated CD3^+^ T cells (**A**) and B cells (**B**) from the patient’s blood sample on days 5, 21, 35, and 75 in comparison to a representative healthy donor (HD). The expression of TCRγδ and CD38 indicates TCRγδ^+^ T cells (**A**) and plasmablasts (**B**), respectively. Highest frequency of TCRγδ^+^ T cells was observed at day 21 and for plasmablasts at day 35, respectively (HD (*n* = 20): %TCRγδ^+^ T cells = 3.1 ± 2.8; %PB = 1).

**Figure 5 vaccines-09-01344-f005:**
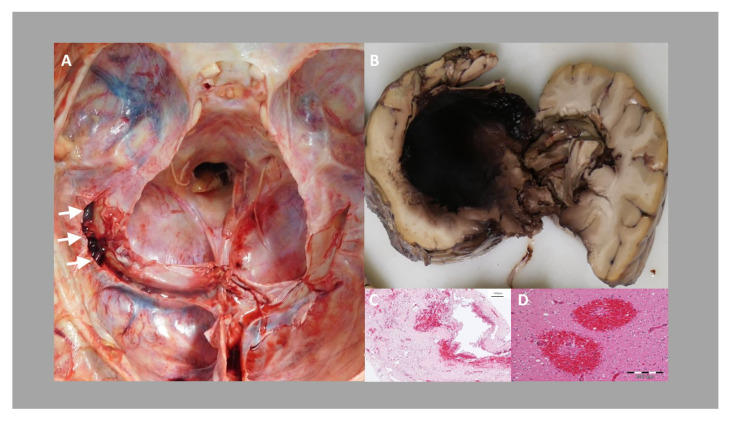
(**A**) Post-mortem diagnosis of left transverse sinus rope-ladder-like organized thrombus (arrows). (**B**) Left-side intracranial hemorrhage with perifocal edema. (**C**) shows fibrous tissue with erythrocyte accumulation comparable with organized thrombus formation. (**D**) Photomicrograph of fresh perivascular hemorrhages in the occipital lobe.

## Data Availability

The datasets used and analyzed during the current study are available from the corresponding author on reasonable request.

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
