# Peer review of "Complicated Long Term Vaccine Induced Thrombotic Immune Thrombocytopenia—A Case Report"

_vaccines, 2021, doi:10.3390/vaccines9111344_

Round 1

Reviewer 1 Report

General comments

In this case report Albrecht Günther and colleagues report on a VIIT reaction after after SARS-CoV-2 vaccination with the Astra Zeneca vaccine. They observed the presence of highly reactive anti-PF4 IgG antibodies over a period of more than 3 months together with cerebral venous sagittal thrombosis and several episodes of intracerebral bleeding (ICB).  

From a principal point of view this is an interesting case report as it shows that highly reactive anti-PF4 IgG antibodies after SARS-CoV-2 vaccination may persist over a longer period of time. However the article does not clarify the question whether these antibodies are causative for the ICBs that led to the patient‘s death. Moreover the authors did not discuss the potential influence of the anticoagulation on the occurrence of ICBs that they observed in their patient.

Specific comments:

It is striking that the authors do not offer any basic information to the patient. There are no specifications regarding the patient’s hight, body weight, BMI, or his medical history regarding diseases that might complicate his clinical course: e.g. hypertension, diabetes, potential cardicac diseases, previous autoimmune diseases and so on.

Line 81

„A 54-year-old Caucasian male presented with new onset of severe headache twelve days after vaccination with ChAdOx1 nCoV-19.“

Was it the 1st or 2nd vaccination?

Line 82

„His prior medical history was unremarkable without evidence for an increased thrombotic risk.“

Was does that mean? See general remarks. Was the patient completely healthy and / or without any medication before vaccination?

Line 96

„IVIG (1g/kg body weight) and anticoagulation with argatroban were started immediately“.

I feel that this is wrong as in Fig. 1A doses of 140 – 160 g ivIgG are presented. Otherwise the patient would have had a body weight of 140 – 160 kg.  

Line 99-103

„Anticoagulation was switched four times because of pharmacologic considerations (liver toxicity, half-life, possibility of antagonization) without affecting the platelet count course (Figure 1B).“

Please add the doses of anticoagulation drugs, in particular the dose of

fondaparinux that was given from day 40 onwards. Did the patient receive fondaparinux until the last day of his life? At which dose?  

The switch of anticoagulation drugs is given in Fig. 1A.

Line 106 – 108

„From thereon, platelets remained >150 x109/L, but still dropped from 581 to 182 x109/L after three months.“

However, the patient’s platelets (PLT) rapidly achieved levels above 400×10*9/L within a few days afterwards. Is there any explanation for this contemporary drop? Was there any change in medication that led to the re-increase of the PLT count?

Line 109 – 112

The authors reported epileptic seizures (line 109) and that he was transferred to a rehab center. How long was the patient in this center? Did he develop his 2nd episode of ICB (day 87 ?) during his stay in the rehab center? If so, was the hospital stay in the rehab center unremarkable (e.g no further seizures, no collapses or other events that might explain the 2nd ICB (day 87 ? ) or his 3rd ICB under anticoagulation (day 103ff)?

Line 114 / 115

„After a period of clinical improvement, four months after VITT on-set the patient developed a new space occupying intracerebral hemorrhage as a secondary complication to previous CVST/ICH and anticoagulation (Figure 2 E&F).“

There is no Figure 2, E&F.

Line 191/192

„In the following weeks, imaging revealed recanalization of the superior sagittal sinus with no further cerebral injury (Figure 2 IV)“.

However, in the subtitle to Figure 2, the authors report the recanalization and a new ICB („a new small ICH in the temporal lobe and the chronic lesions (arrows)“) for day 87 (Fig. 2, IV and V). This is a clear contradiction to line 191-192.

Line 234 – 236

„The complicated long-term course of the present case has to be seen in conjunction with high titers of platelet activating anti-PF4 IgG persisting for longer than three months.

The authors did not show that thgere were high titers of anti-PF4 IgG, but they have shown that there were highly reactive anti-PF4 IgG. But nevertheless as a reviewer I agree that the long-term presence of highly reactive anti-PF4 IgG is a remarkable finding. However, this can be also considered as an immunologic phenomenon without further clinical relevance as the patient did not demonstrate further episodes of thromboses. Moreover, the repeated ICB (day 87 ? and day 103) in this patient may be the consequence of long – term anticoagulization only.

Line 253 – 254

„In our patient, thrombocytopenia recovered significantly after starting prednisolone, a treatment which was initiated to decrease the production of PF4.

When I look on Fig. 1A, I do not have the impression that prednisolone was started to correct the PLT count but to reduce the levels or reactivity of anti-PF4 IgG.

Author Response

Reviewer comments and answer to the reviewers

We first want to thank both reviewers for their helpful comments and we hope by revising the manuscript according to the reviewers comments the case report is now suitable for publication in Vaccines.

Reviewer 1:

In this case report Albrecht Günther and colleagues report on a VIIT reaction after after SARS-CoV-2 vaccination with the Astra Zeneca vaccine. They observed the presence of highly reactive anti-PF4 IgG antibodies over a period of more than 3 months together with cerebral venous sagittal thrombosis and several episodes of intracerebral bleeding (ICB).  

From a principal point of view this is an interesting case report as it shows that highly reactive anti-PF4 IgG antibodies after SARS-CoV-2 vaccination may persist over a longer period of time. However the article does not clarify the question whether these antibodies are causative for the ICBs that led to the patient‘s death. Moreover the authors did not discuss the potential influence of the anticoagulation on the occurrence of ICBs that they observed in their patient.

Specific comments:

It is striking that the authors do not offer any basic information to the patient. There are no specifications regarding the patient’s hight, body weight, BMI, or his medical history regarding diseases that might complicate his clinical course: e.g. hypertension, diabetes, potential cardiac diseases, previous autoimmune diseases and so on.

Line 81

„A 54-year-old Caucasian male presented with new onset of severe headache twelve days after vaccination with ChAdOx1 nCoV-19.“

Was it the 1st or 2nd vaccination?

The VITT manifestation occurred after the first vaccination. We corrected the text accordingly (lane 88)

„A 54-year-old Caucasian male presented with new onset of severe headache twelve days after the first vaccination with ChAdOx1 nCoV-19.“

Line 82

„His prior medical history was unremarkable without evidence for an increased thrombotic risk.“

Was does that mean? See general remarks. Was the patient completely healthy and / or without any medication before vaccination?

The patient was completely healthy without previous medical events apart from spontaneous pneumothorax events in 2010 which is in the authors view unremarkable in the clinical context. He was on no regular medication.  

His prior medical history was unremarkable without evidence for an increased thrombotic risk. The patient did not receive any regular medication “ (lane 90)

Line 96

„IVIG (1g/kg body weight) and anticoagulation with argatroban were started immediately“.

I feel that this is wrong as in Fig. 1A doses of 140 – 160 g ivIgG are presented. Otherwise the patient would have had a body weight of 140 – 160 kg.  

This is misunderstandable: with an approximate body weight of 70-75 kg bodyweight he received app. 1g/kgKG on two consecutive days (we corrected the text accordingly).

IVIG (1g/kg body weight at two consecutive days each  and anticoagulation with argatroban were started immediately (Lane 103; lane 135&136 in Figure legend 1)

Line 99-103

„Anticoagulation was switched four times because of pharmacologic considerations (liver toxicity, half-life, possibility of antagonization) without affecting the platelet count course (Figure 1B).“

Please add the doses of anticoagulation drugs, in particular the dose of

fondaparinux that was given from day 40 onwards. Did the patient receive fondaparinux until the last day of his life? At which dose?  

The switch of anticoagulation drugs is given in Fig. 1A.

We now added the dosage of the administered anticoagulative drugs accordingly in the legend of Figure 1A (lane 136-140):

Argatroban: aPTT controlled administration with changing dosage (aiming at a PTT of 60 sec.; app. 2µg/kg/min)

Tinzaparin: 12.000 U / d

Danaparoid: 3x750 Units s.c./d according to anti-Xa-level

Fondaparinux:  7.5 mg/d s.c.

Fondaparinux was administered with the dosage of 7.5 mg/d until the treatment goal was changed to best supportive care after diagnosis of recurrent ICH.

Line 106 – 108

„From thereon, platelets remained >150 x109/L, but still dropped from 581 to 182 x109/L after three months.“

However, the patient’s platelets (PLT) rapidly achieved levels above 400×10*9/L within a few days afterwards. Is there any explanation for this contemporary drop? Was there any change in medication that led to the re-increase of the PLT count?

The only change which could partially influence the PLT count at this point was the decreased prednisolone dosage before the PLT count rapidly decreased. However, PLT count increased without another change in medication, including prednisolone at this point. No other clear co-morbidities or other potential pathological events were clearly associated with this PLT drop apart from still very high PF4-antibody levels. We have added this information in the text.  (Lane 115-116)

From thereon, platelets remained >150 x109/L, but still dropped from 581 to 182 x109/L after three months, which could not be explained with changes in medication or other new co-morbidities.

Line 109 – 112

The authors reported epileptic seizures (line 109) and that he was transferred to a rehab center. How long was the patient in this center? Did he develop his 2nd episode of ICB (day 87 ?) during his stay in the rehab center? If so, was the hospital stay in the rehab center unremarkable (e.g no further seizures, no collapses or other events that might explain the 2nd ICB (day 87 ? ) or his 3rd ICB under anticoagulation (day 103ff)?

The patient was sent to the rehab centre after 40 days and returned to our hospital after six weeks for diagnostic and therapeutic re-evaluation. The minor re-ICH was a diagnosed in routine imaging control without a clinical correlate at this time point. He was sent to another rehab center at day 90 for further neurological rehabilitation. At day 103 clinical signs for increased ICP developed with vomitting, decreased level of consciousness, pupillary abnormalities and new swelling at the site of hemicraniectomy as a sign of herniation. We have added missing information in the text (lane 119-127):

The patient was transferred to a neurological rehabilitation centre after 40 days and returned to our hospital after six weeks for diagnostic and therapeutic re-evaluation. The minor Re-ICH (Figure 2V) was diagnosed in a routine imaging control without a clinical correlate at this time point. He was sent to another rehabilitation center at day 90 for further neurological rehabilitation. At day 103 he developed new clinical signs for increased ICP  with vomitting, decreased level of consciousness, pupillary abnormalities and clinical swelling at the site of hemicraniectomy as a sign of herniation.

Line 114 / 115

„After a period of clinical improvement, four months after VITT on-set the patient developed a new space occupying intracerebral hemorrhage as a secondary complication to previous CVST/ICH and anticoagulation (Figure 2 E&F).“

There is no Figure 2, E&F.

We corrected the sentence (lane127-130) :

Four months after VITT onset the patient developed a new space occupying intracerebral hemorrhage as a secondary complication to previous CVST/ICH and anticoagulation (Figure 2 V&VI).“

Line 191/192

„In the following weeks, imaging revealed recanalization of the superior sagittal sinus with no further cerebral injury (Figure 2 IV)“.

However, in the subtitle to Figure 2, the authors report the recanalization and a new ICB („a new small ICH in the temporal lobe and the chronic lesions (arrows)“) for day 87 (Fig. 2, IV and V). This is a clear contradiction to line 191-192.

We have corrected the text accordingly:

MRI after decompression hemicraniectomy revealed no new pathology (Figure 2 III; right frontal ICH was detected on pre-operative CT scan already). Also, during the following weeks, imaging showed recanalization of the superior sagittal sinus with no further cerebral injury (Figure 2 IV). (lane 156-157)

We also changed the Figure legend accordingly:

Lane 124: MRI (fluid attenuated inversion recovery; FLAIR) 3 weeks after decompression hemicraniectomy showed the known bifrontal hemorrhagic infarction (#). (Lane 144-145)

Line 234 – 236

„The complicated long-term course of the present case has to be seen in conjunction with high titers of platelet activating anti-PF4 IgG persisting for longer than three months.

The authors did not show that there were high titers of anti-PF4 IgG, but they have shown that there were highly reactive anti-PF4 IgG. But nevertheless as a reviewer I agree that the long-term presence of highly reactive anti-PF4 IgG is a remarkable finding. However, this can be also considered as an immunologic phenomenon without further clinical relevance as the patient did not demonstrate further episodes of thromboses. Moreover, the repeated ICB (day 87 ? and day 103) in this patient may be the consequence of long – term anticoagulation only.

Thank you for this interesting discussion. High titers of anti PF4-IgG platelet activating antibodies are associated with a prothrombotic condition in VITT. Therapeutic dose anticoagulation can resolve this and reduce the risk of thrombosis. This is indicated by the decrease of D-Dimer levels towards the later course. Based on our experience, VITT patients on therapeutic dose anticoagulation rarely develop new thromboses, but it is likely that the prothrombotic state returns, when anticoagulation is stopped albeit platelet activating antibodies remain at high titers, as transferred from our experience with autoimmune heparin induced thrombocytopenia. This provides the rationale to continue anticoagulation, as long as these antibodies are present. However, no clinical study to date is available supporting this strategy in VITT and we agree indeed with this reviewer, that the clinical relevance is questionable for this reason. We see the ICH clearly as a complication from lesions acquired early in VITT-related CSVT and secondary to anticoagulation and not as a result from the high titre PF4 antibodies. We clarified this (see below)

 Line 296-298

Thrombocytopenia recovered significantly after starting prednisolone, a treatment which was initiated to decrease the production of PF4”.

When I look on Fig. 1A, I do not have the impression that prednisolone was started to correct the PLT count but to reduce the levels or reactivity of anti-PF4 IgG.

Thank you also for this statement. We explained the recurrent episodes of thrombocytopenia by the high titer PF4-antibodies, resulting in recurrent platelet activation and platelet consumption. This was supported by the concomitant increase of D-Dimers. We started immunosuppression in order to reduce PF4 antibody levels with the aim to control the platelet counts.

We clarified this together with the response of the previous comment (lane 267-279):

During the early course, recurrent episodes of thrombocytopenia developed as a consequence of platelet activation and platelet consumption. This was sup-ported by the concomitant increase of D-Dimers during recurrent thrombocy-topenia. High titers of anti-PF4-IgG platelet activating antibodies promote the prothrombotic condition in VITT and therapeutic dose anticoagulation is nec-essary to reduce the risk of thrombosis. We consider it likely that the pro-thrombotic state returns, when anticoagulation is stopped albeit platelet acti-vating antibodies remain at high titers, based on our experience with autoimmune heparin induced thrombocytopenia [23]. However, the ICH developed as a complication from lesions acquired early in VITT-related CSVT and secondary to anticoagulation. It is therefore important to define the optimal time point and strategy to stop or reduce the dose of anticoagulation in VITT (e.g. by close monitoring of platelet counts, D-Dimers, and Anti-PF4-IgG platelet activating antibodies).  

Reviewer 2 Report

Dear authors, I congratulate you for conducting the present case report. Here goes a few minor considerations:

TITLE

Since the study design is a “Case Report” that terminology should be included in the Title.

ABSTRACT

Please define IVIG abbreviation. Is it Intravenous Immunoglobulin TherapY?

KEYWORDS

The term “Case Report” should be included in the keywords

CASE REPORT

I see the authors have described the clinical evolution under the “Case Report” subheading and then several other complementary examination procedures under other subheadings. All this subheading are related to the Case Report description itself. So my recommendation would be to add a subheading after the Case Report heading, for instance “Clinical evolution” (or similar, that the authors fell suitable) and later, instead of Methods, add something such as “Labotarorial testing, Imaging and complementary examinations” and describe, under that subheading the testing that were conducted, and later their results. The way it is presented look somehow weird. The Case Report usually have procedures, but not the “Material & Methods” and “Results” such as in other study design. So my recommendation is to add this subheading a better orientation to the Case Report presentation format.

DISCUSSION

Please do not start with “Our patient…”, these personal pronouns should be avoid.

Author Response

Reviewer comments and answer to the reviewers

2, Reviewer: 

Dear authors, I congratulate you for conducting the present case report. Here goes a few minor considerations:

TITLE

Since the study design is a “Case Report” that terminology should be included in the Title.

We now included „Case report“ in the title: Complicated long term vaccine induced thrombotic immune thrombocytopenia – a case report. (Lane 3)

ABSTRACT

Please define IVIG abbreviation. Is it Intravenous Immunoglobulin Therapy?

IVIG: intravenous immune globuline

The is identifiable as a treatment as in the present version: ….Repeated treatment with intravenous immune globuline (Lane 35-36).

KEYWORDS

The term “Case Report” should be included in the keywords

We included „Case report“ in the keywords.(Lane 44)

CASE REPORT

I see the authors have described the clinical evolution under the “Case Report” subheading and then several other complementary examination procedures under other subheadings. All this subheading are related to the Case Report description itself. So my recommendation would be to add a subheading after the Case Report heading, for instance “Clinical evolution” (or similar, that the authors fell suitable) and later, instead of Methods, add something such as “Labotarorial testing, Imaging and complementary examinations” and describe, under that subheading the testing that were conducted, and later their results. The way it is presented look somehow weird. The Case Report usually have procedures, but not the “Material & Methods” and “Results” such as in other study design. So my recommendation is to add this subheading a better orientation to the Case Report presentation format.

Thank you for this helpful comment. We have arranged the report according to the reviewers comments.

DISCUSSION

Please do not start with “Our patient…”, these personal pronouns should be avoid.

This has been changed accordingly. : The patient presented…. (Lane 257)

Furthermore, we have added neuropathological findings (243-246) in the text as well as in Figure 5, which has been re-arranged (lane 250-254). These findings in our opinion complete this case report with additional visualization of this fatal VITT case.

Round 2

Reviewer 1 Report

None.

Reviewer 2 Report

Dear authors, I have no more concerns. Thank you

This manuscript is a resubmission of an earlier submission. The following is a list of the peer review reports and author responses from that submission.